# Analysis of Inverter Efficiency Using Photovoltaic Power Generation Element Parameters

**DOI:** 10.3390/s24196390

**Published:** 2024-10-02

**Authors:** Su-Chang Lim, Byung-Gyu Kim, Jong-Chan Kim

**Affiliations:** 1Department of Computer Engineering, Sunchon National University, Suncheon 57992, Republic of Korea; suchanglim@scnu.ac.kr; 2Division of AI Engineering, Sookmyung Women’s University, Seoul 04310, Republic of Korea

**Keywords:** PV system, PV power forecasting, AI, data analysis, deep learning, LSTM

## Abstract

Photovoltaic power generation is influenced not only by variable environmental factors, such as solar radiation, temperature, and humidity, but also by the condition of equipment, including solar modules and inverters. In order to preserve energy production, it is essential to maintain and operate the equipment in optimal condition, which makes it crucial to determine the condition of the equipment in advance. This paper proposes a method of determining a degradation of efficiency by focusing on photovoltaic equipment, especially inverters, using LSTM (Long Short-Term Memory) for maintenance. The deterioration in the efficiency of the inverter is set based on the power generation predicted through the LSTM model. To this end, a correlation analysis and a linear analysis were performed between the power generation data collected at the power plant to learn the power generation prediction model and the data collected by the environmental sensor. With this analysis, a model was trained using solar radiation data and power data that are highly correlated with power generation. The results of the evaluation of the model’s performance show that it achieves a MAPE of 7.36, an RMSE of 27.91, a MAE of 18.43, and an R2 of 0.97. The verified model is applied to the power generation data of the selected inverters for the years 2020, 2021, and 2022. Through statistical analysis, it was determined that the error rate in 2022, the third year of its operation, increased by 159.55W on average from the error rate of the power generation forecast in 2020, the first year of operation. This indicates a 0.75% decrease in the inverter’s efficiency compared to the inverter’s power generation capacity. Therefore, it is judged that it can be applied effectively to analyses of inverter efficiency in the operation of photovoltaic plants.

## 1. Introduction

Energy is one of the key factors that enhance industry and productivity [1]. As energy consumption increases, the use of fossil fuels also increases, and environmental pollution increases accordingly. In response to the global climate change crisis, the transition from traditional fossil fuels to cleaner and more energy-efficient renewable energy sources is being accelerated [2,3]. Among various renewable energies, the installation of photovoltaic power generation continues to expand. Photovoltaic power, an abundant and clean resource, is one of the alternative forms of energy used to reduce greenhouse gas emissions. With the growing concerns about environmental pollution and energy shortages, photovoltaic has aroused expectations that it will play a significant role in overcoming energy crises by supplying electricity to future societies [4]. In the Republic of Korea, which is following the global trend, the government and the private sector have also led the way and strive to expand and distribute new and renewable energy. In particular, the government has implemented the “Green New Deal” policy to expand and distribute photovoltaic and wind power. As a part of this policy, the goal for the widespread adoption of photovoltaic and wind power has been raised to 42.7 GW by the year 2050 [5]. People who have developed an interest in photovoltaic power plants through renewable energy adoption policies have naturally begun to focus on methods of maintaining these plants in optimal condition.

The key equipment to consider regarding the lifespan of a photovoltaic power generation system is its solar modules and inverters. Solar modules are essential devices that convert photovoltaic energy into electrical energy; however, they degrade over time due to exposure to ultraviolet radiation. In the case of solar modules, strings of modules are formed by connecting them in series or in parallel for the purpose of power generation. But a failure or rapid decrease in the efficiency of some modules in a string will reduce the overall efficiency of the entire photovoltaic power generation system, adversely affecting the total amount of power generated. A solar inverter is a device that converts the DC (direct current) produced by solar modules into AC (alternating current). Generally, not only solar modules but also system equipment, including inverters, continue to deteriorate in term of performance, eventually reducing power generation [6]. Therefore, maintaining a photovoltaic power plant in its optimal operational state makes it possible to manage potential risks while preserving power generation efficiency, thus enhancing the reliability of a solar power system.

It is essential to have criteria for analyzing the anomaly phenomenon observed in photovoltaic power generation systems in order to maintain the efficiency of a power plant [7]. One of the key criteria for analyzing the efficiency of a photovoltaic power plant is the currently generated amount of power [8]. Photovoltaic power generation is closely influenced by weather and other environmental factors. Though the generated power varies unpredictably with weather conditions, if one can predict power generation based on meteorological conditions, it becomes possible to analyze efficiency by analyzing fluctuations in total power generation. Based on this power generation, the diagnosis of the state of a solar module and an inverter should be carried out.

Predicting photovoltaic power generation in photovoltaic systems requires a deep understanding of current status data. The amount of power generated in a given plant is variable because it is affected by difficult-to-control variables, such as season, time, plant capacity, and weather conditions [9]. All the data available from a plant consists of time series data. Time series models are particularly useful when it is difficult to measure the relationship between explanatory variables that explain data fluctuations. Predicting factors that influence photovoltaic power generation, such as solar radiation or solar panel output, is important in terms of ensuring a smooth linkage between smart grids and PV systems [10,11]. In addition to the solar radiation and the solar panel’s output, many factors influence the amount of photovoltaic power generation, and it is not easy to clearly define the relationships between such factors. In this case, time series models can be used effectively in research aimed at predicting photovoltaic power generation [12,13,14]. Data preprocessing is essential when conducting research using time series models. Noise data that may occur during data collection may also introduce bias into a prediction model, which in turn distorts the results of data analysis. To obtain accurate analytical results, it is necessary to process noise data by considering the characteristics of the entire dataset.

The main objective of this study is to evaluate the efficiency, performance, and reliability of a photovoltaic power generation system. The evaluation was conducted by analyzing three years of power generation data, i.e., the daily and monthly energy production data of an inverter, which were obtained by a data collection device. By applying preprocessing to each element’s data, the form of the data was refined into an easy form for analysis by removing any factors that might affect the analysis of noises and missing values. Next, the refined data were subjected to a correlation analysis to derive essential data for the efficiency analysis, and a power generation prediction model was constructed using a Long Short-Term Memory (LSTM) designed for time series data analysis and prediction. The power generation prediction model is verified through quantitative evaluation metrics and statistical analysis. The model is used to predict the power generation of old inverters and to analyze the difference between the prediction result and the actual amount of power generation by using statistical analysis techniques.

The composition of this paper is as follows: Section 2 describes the study on the power generation prediction algorithm using an antecedent time series data analysis algorithm; Section 3 describes the process of deriving correlations through data preprocessing and analysis; Section 4 presents the power generation prediction results and the results of the analysis of inverter efficiency derived using quantitative and qualitative indicators with the proposed predictive model; and, finally, Section 5 presents the conclusion.

## 2. Related Works

The inverter, the main component of photovoltaic power generation systems, is an item of power generation equipment that converts electricity generated by solar modules from DC to AC. Inverter power generation data are connected to the data collection device and collected through the inverter’s unique protocol communication. In cases where environmental sensors are installed in a photovoltaic power generation system, data on external temperature, module temperature, slope solar radiation, and horizontal solar radiation data are also collected. Data acquired at power plants, such as power generation, solar radiation, temperature, and such like, are single time series data. Conducting only a simple comparison of such data cannot predict the state of power generation [15]. Previous studies have used various predictor variables to predict power generation and diagnose issues based on errors between predictions and actual values. Recently, research has been conducted to diagnose failures in photovoltaic power generation systems, including panels and other facilities, using electrical characteristics, MPPT, solar radiation, precipitation, temperature, total cloud cover, humidity, and fog data [16,17,18].

The methods used to predict power generation are mainly statistical models and machine learning (ML) models. A statistical model is a method of solving problems by deriving characteristics and correlations between past data and current data [19]. The performance of this model is determined by the modeling completeness that expresses the relationship between the collected power generation data and the environmental variables. To achieve this, it requires a sufficient data range for high-quality modeling as well as consistent, high-quality data that do not include noise and the like. Common statistical models include the RM (Regression Method), the ARMA (Autoregressive Moving Average), the ARIMA (Autoregressive Integrated Moving Average), the KF (Kalman Filter), and the SVM (Support Vector Machine).

In particular, the ARMA (Autoregressive Moving Average) is a model that combines two models to analyze data that cannot be fully explained by the AR (Autoregressive) model or the MA (Moving Average) model. However, the ARMA has a limitation in that it can only be applied to time series data with constant mean and variance, which does not change over time. As for the ARIMA, it is a model that adds a differencing process to the ARMA model and is used for non-stationary time series, taking into consideration seasonality. A study that classifies them into specific environments by combining the SVM and the Random Forest and predicts power generation was conducted [20]. The SVM algorithm was used to predict power generation, and the Random Forest combined predicted values and analyzed them in conjunction with the SVM results. In this study, past and present power generation data and weather data were mixed and used as the input. Research was conducted on which data, including weather condition data, were classified into specific environments using the SVM, and a corresponding model for each environment was used to predict power generation [21]. However, statistical methods rely solely on past data and the model, which show limitations in their predictive performance due to the non-stationary characteristics inherent in photovoltaic time series data [22].

Research was conducted to predict unit power generation and diagnose the failures of photovoltaic power generation systems using the SVR (Support Vector Regression), ANN (Artificial Neural Networks), and Deep Learning [23,24,25]. In particular, ANN are a widely used method for prediction tasks due to MLP (Multi-Layer Perceptron), which consists of fully connected nodes. As the input data pass through a layer composed of weighted values, ANN are trained to be optimized to extract specific patterns. ANN, through a form of non-linear modeling, outperform traditional statistical methods. As a result, they can perform non-linear mapping from input to output through error feedback. Due to these characteristics, ANN-based research was conducted to predict solar radiation in real time. In addition, research was conducted by utilizing hybrid methods that combine ANN and the Random Forest, local temperature, and sunshine duration [26,27].

In recent times, research has been conducted to apply deep learning to time series fields such as power generation prediction [28]. Unlike the ML algorithm, deep learning consists of multiple stacked layers. It can extract non-linear correlations from complex datasets and learn how to extract features on its own without requiring separate feature extraction algorithms [29]. The RNN (Recurrent Neural Network) is a type of artificial neural network designed for time series data [30]. The RNN has a feedback loop in the recurrent layer that stores the “past time point” information of time series data in its memory. However, there is a problem in that historical information is lost if the timing of the data is prolonged.

Recently, LSTM (Long Short-Term Memory) has become one of the key artificial neural networks used to predict photovoltaic power generation [31]. LSTM outperforms MLP (Multi-Layer Perceptron) and other traditional methods, and it has become a superior and more efficient predictive algorithm. LSTM-RNN was proposed to predict the output power of photovoltaic power generation systems more accurately using hourly datasets for one year, and it showed lower prediction errors than other predicting algorithms [32]. Research using LSTM to predict solar panel power generation in the medium and long term has been conducted, showing lower error rates compared to the SVM [33]. To improve the PV prediction value, a hybrid model consisting of a structure in which a drop-out layer and an LSTM layer are repeatedly stacked on a multistep CNN was proposed [34].

Forecasting methodologies can be broadly classified into short-term, medium-term, and long-term forecasting. Short-term forecasting typically covers periods of one hour or less, medium-term forecasting spans up to a day, and long-term forecasting extends from several days to over a week [35]. These approaches can be categorized into physical, statistical, and hybrid methods. Physical methods focus on predicting solar power generation based on numerical weather prediction (NWP) models and the physical principles of PV cells [36]. Unlike statistical methods, physical methods do not require historical data but instead rely on the geographical characteristics of PV installations and detailed meteorological inputs. Statistical methods, on the other hand, use historical data to establish a relationship between past time series data and solar energy output. The performance of statistical models depends on how well they map past observations of power generation to corresponding weather and climate parameters. Generally, statistical methods involve simpler modeling processes compared to physical methods.

## 3. Proposed Method

### 3.1. Data Preparation and Data Shape Analysis

This study was conducted on inverters from the same manufacturer. Photovoltaic power plants are equipped with inverters and environmental sensors, and power generation data and environmental data are collected at one-minute intervals. The inverter started generating power in March 2020. The average annual photovoltaic power generation time in the region is 3.5 h, the average wind speed is 4~5 m/s, and the meridian altitude is 53° in spring (March–May), 76.5° in summer (June–August), 53° in autumn (September–November), and 29.5° in winter (December–February). Table 1 and Table 2 show the collected DC power generation, AC power generation, and environmental sensor data in schematic form.

In Table 1, ENV_DATE and ENV_TIME represent the date and time when the data were collected, while ENV_Slopesolar and ENV_Levelsolar are the solar radiation data. The last ENV_Modetemp represents the temperature of the solar modules, and ENV_Airtemp refers to the external temperature. In Table 2, POW_DATE and POW_TIME indicate the date and time when the inverter data were collected. In Table 2, rows 4 to 9 represent the power generation data.

The data used for prediction were processed by dividing them into 15-min intervals for analysis. Missing values in time series data may cause the time intervals of the data to become uneven, so they were preprocessed using moving averages. Power generation data and environmental sensor data were stored in separate spaces, so power generation and solar radiation data were matched in pairs. Issues with the inverter or data collection device can result in missing values in the collected data, which can in turn affect the temporal sequence of the data. Also, abnormal elements in the input data for the prediction model are likely to result in high prediction errors. Therefore, inadequate training issues and computational costs can be reduced by preprocessing the input data, thus improving the model’s accuracy. The proposed model predicts future data using continuous time series data, so uniform performance of the model should be guaranteed. Therefore, a time interpolation was applied using the values of x(t − 1) and x(t + 1) of the time when missing data occurred. The missing data interpolation method is as follows. Among the matched datasets, data for which sensor data were not entered for more than 10 min were considered to be missing values and removed, whereas data entered for less than 10 min were processed using a moving average algorithm in order to refine the data.

Before creating the model, the data to be used in the model were selected through regression analysis in order to confirm the decrease in the inverter’s efficiency. A highly accurate power generation prediction model was required to diagnose the exact value of how much the inverter’s efficiency had decreased. First, the results of an analysis of the elementary statistics of the data conducted as a pre-verification step in fabricating the model are shown in Table 3.

In Table 3, which shows the elementary statistics, the median of the DC and AC outputs is smaller than their mean, but their standard deviations are 5303 and 5097.8, respectively, showing a skewed left shape close to the normal distribution, where the difference between the median and the mean is not large. Furthermore, to improve the prediction accuracy of the model, a correlation analysis was conducted to examine the relationships between the collected data at the solar power plant. A value with a high correlation coefficient was selected, with the AC power generation to be predicted as the input value. As a result of the correlation analysis, among the inverter data, the value with the highest correlation coefficient with AC power generation was DC power generation. As the converted value of DC power is AC power, the relationship between these two can be considered to be directly related to the inverter’s efficiency. The correlation coefficient is a numerical value representing strength according to the relationship between the two variables. There are various methods of calculating correlation coefficients, but the Pearson correlation coefficient is commonly used. The covariance between the two variables is defined as the value divided by the product of the standard deviations of the two variables, as shown in Equation (1):(1)$$\rho \left(X,Y\right)=\frac{COV(X,Y)}{\sigma_{X}\sigma_{Y}}.$$

Here, the covariance $COV\left(X,Y\right)$ between X and Y is additionally defined as the expected value of the product of the deviations of x and y from their respective means. This can be made even clearer by using the covariance formula in Equation (2), as follows:(2)$$COV\left(X,Y\right)=E[\left(X-\mu_{X}\right)\left(Y-\mu_{Y}\right)].$$

Therefore, the final formula for Pearson’s correlation coefficient can be summarized as Equation (3):(3)$$\rho \left(X,Y\right)=\frac{E[\left(X-\mu_{X}\right)\left(Y-\mu_{Y}\right)]}{\sigma_{X}\sigma_{Y}}$$

Here, the ρ(rho) exists between −1 and +1. A value close to +1 indicates a strong positive relationship between x and y, while a value close to −1 indicates a strong negative relationship between *X* and *Y*. However, a value close to 0 means there is no significant relationship between x and y.

Figure 1 shows a correlation matrix analyzing the correlation between the power generation data. The darker areas indicate a low correlation, while the brighter areas indicate a high correlation. The degree of correlation ranges from −1 to 1. DCP shows that the correlation value with DC is 0.99, indicating a close relationship between them. Generally, when DC power generation is high, AC power generation is also high, which is why the correlation coefficient is high. Therefore, it confirms that the correlation coefficient between AC and DC power generation is one in the data.

Next, a linear regression analysis was conducted to determine the input data for implementing the power generation prediction model. It was necessary to improve the accuracy of the power generation prediction model in order to diagnose the decrease in the inverter’s efficiency. Since the accuracy of the model depends on the types of input data, it is important to select meaningful data to obtain significant results. The input data were selected based on the high correlation with the AC power generation to be predicted. The results of the correlation analysis showed that, among the inverter data, the value with the highest correlation coefficient with AC power generation was DC power generation. Since DC power generation is converted into AC power generation, the relationship between these two can be considered to be directly related to the inverter’s efficiency. In addition, the data with the highest correlation coefficient among the environmental sensor data for controlling the difference in the inverter installation position and the weather variables were the data on the amount of solar radiation.

Figure 2 shows the scatter plots of the linear analysis of each input data. Figure 2a,b show the linear relationships of DC output with the level and slope of solar radiation. The correlation coefficient between the level solar radiation and the DC output is 0.9257, while the correlation coefficient between the slope solar radiation and the DC output is 0.9762, indicating a positive correlation between them. Figure 2c,d illustrate the linear relationships of the AC output with the level and slope of solar radiation. The correlation coefficient between the level solar radiation and the AC output is 0.9781, while the correlation coefficient between the slope solar radiation and the AC output is 0.9276, indicating a positive correlation between them. Figure 2e shows the linear relationship between the DC output and the AC output, with a correlation coefficient of 0.99. In the case of Figure 2e, AC is converted from DC, leading to a high correlation. The thickness of the dots in the scatter plot visually represents the magnitude of the data values. These data values correspond to numerical values that reflect either the output or the correlation between variables. A larger dot indicates a higher data value or a stronger correlation at that point, while a smaller dot indicates a lower value. This visualization enables a more intuitive understanding of the distribution of the data and the strength of the correlations. Specifically, in the context of a solar power generation system, the greater the power output, the thicker the corresponding dot. Conversely, a lower output value is represented by a thinner dot. In this result, the DC output correlates more strongly with the AC output than solar radiation, and the density is widely distributed. Each of these correlations is summarized in Table 4.

The distribution pattern of the solar radiation data varies significantly depending on the data collection period. Such characteristics can have an impact on the model’s prediction performance. Therefore, to maximize prediction performance, an attempt was made to control the variables as much as possible by using data from common power generation timeframes between December and January. As shown in Table 3, the DC power output ranges from a minimum of 660 to a maximum of 22,500; the AC power output ranges from a minimum of 653 to a maximum of 21,512.5, and the temperature ranges from a minimum of -18.1 degrees to a maximum of 62.15 degrees. In the case of solar radiation, the data range is from a minimum of 17.2 to a maximum of 1088.5. Since the scales for each type of data are different, it is necessary to prevent the model from assigning more importance to the larger data by converting the data shape to a similar scale. Also, the different scales of the data can have a disproportionate effect on the model’s learning performance. Therefore, normalization was performed to ensure that the model is trained in a balanced manner by limiting the impact of the outliers in specific data. The MinMaxScaler algorithm was applied to all data pairs used in the study, as shown in Equation (4), and the data were normalized within the range of −1.0 to 1.0.
(4)$$z=\frac{x-x_{min}}{x_{max}-x_{min}}$$

Here, x is the actual data, $x_{min}$ is the minimum value, $x_{max}$ is the maximum value, and z represents the normalized value.

### 3.2. Model Creation and Verification

The LSTM is an RNN-based architecture that has been shown to perform well in processing time series data. The existing RNN (Recurring Neural Network) is trained by disseminating information based on data from past time points. A vanishing gradient problem, wherein the gradient decreases during the training process, may occur. Additionally, the RNN can cause the “gradient exploding” problem in which the gradients increase rapidly during training, causing the weight values to become abnormally large and resulting in divergence. As such, the LSTM is an architecture that solves these issues inherent to traditional the RNNs, and it is also highly suited to tasks like time series prediction as it can capture long-range dependencies within a sequence. Unlike the RNN, the LSTM incorporates memory cells and various gates that control the information flow, allowing it to maintain information over longer sequences without losing information about past data. Figure 3 illustrates the structure of the LSTM used in this paper to predict the AC output.

The LSTM receives the three types of data at each time step. A short-term memory called the “hidden state”, a long-term memory called the “cell state”, and the current data are used by the LSTM. In Figure 3, the short-term memory is denoted as $h_{t}$, the long-term memory as $C_{t}$, and the current data as $x_{t}$. In an LSTM, the cell is an element used to store and manage information on data at past time points and current time points in time series data. The cell uses gates to perform filtering based on importance in order to transfer long-term and short-term information to the next cell. Each cell operates through an input gate, a forget gate, and an output gate, and each gate controls the information flow inside and outside the cell.

In each equation, the weights w are important parameters that define the relationship between the current input and the previous hidden state. The weight matrices are stored as trainable parameters of the LSTM model. These matrices are updated during the training process of the model and determine how much information each gate takes from the input, retains from the previous hidden state, and passes on to the new hidden state. This can be explained as follows: $w_{f},\;w_{i},\;w_{c},\;{and\;w}_{o}$ are weight matrices for the current input data $x_{t}$, and ${wh}_{f},\;{wh}_{i},\;{wh}_{c},\;{and\;wh}_{o}$ represent weight matrices for the previous hidden state $h_{t-1}$.

The input gate determines what new information needs to be stored in the cell state. It does this by considering the current input and the previous cell state. The gate operation converts each datum in the cell state to a value between 0 and 1 using the sigmoid function, one of the activation functions. Equation (5) below shows the process of the input gate as a mathematical formula:(5)$$i_{t}=\sigma \left(x_{t}w_{i}+h_{t-1}{wh}_{i}+b_{i}\right).$$

Here, $i_{t}$ is the input gate vector, $w_{i}$ is the weight matrix applied to the input gate, $x_{t}$ is the input data at the current time, $h_{t-1}$ is the hidden state output at the previous time point, ${wh}_{i}$ is the weight matrix for hidden state, and $b_{i}$ is a learnable parameter for maintaining model nonlinearity.

The forget gate determines which information is to be removed by identifying information retained or unnecessary in the previous cell state. It is performed through the sigmoid function in the same way as the input gate. The closer to 0, the more information about the previous state is lost, and the gate decides which data are to be omitted. Values close to 1 mean that the previous state’s data will be retained and transferred to the next gate. While the data go through the forget gate, they pass the process of selecting information to be stored. The forget gate discards information from the previous time point and adds new information to be memorized. This newly added information determines the values of each element. Equation (6) below represents the forget gate process in a mathematical formula:(6)$$f_{t}=\sigma \left(x_{t}w_{f}+h_{t-1}{wh}_{f}+b_{f}\right).$$

Here, $f_{t}$ is the forget gate vector, $w_{f}$ is the weight matrix applied to the forget gate, $x_{t}$ represents the input data at the current time, $h_{t-1}$ is the hidden state from the previous time point, ${wh}_{f}$ is the weight matrix for the hidden state, and *b_f_* is the bias used in the forget gate.

Equation (7) below represents the formula for determining the new candidate cell state. It is calculated based on the current input data and the previous hidden state:(7)$$g_{t}=tanh\left(x_{t}w_{c}+h_{t-1}{wh}_{c}+b_{c}\right).$$

Here, $g_{t}$ represents the candidate cell state, $w_{c}$ is the weight matrix for the candidate cell, ${wh}_{c}$ is the weight matrix for the hidden state, and $b_{c}$ indicates the bias used in the forgot gate.

Equation (8) below represents the formula for updating the new cell state by combining information from the forget gate, the input gate, and the candidate cell state:(8)$$c_{t}=f_{t}⨀c_{t-1}+g_{t}⨀i_{t}.$$

Here, $c_{t}$ represents the updated cell state at the current time step, $f_{t}$ is the output of the forget gate, $c_{t-1}$ is the cell state from the previous time step, and $i_{t}$ is the output of the input gate. The term $g_{t}$ represents the candidate cell state, and ⊙ indicates element-wise multiplication.

Equation (9) below represents the formula for updating the new hidden state by combining information from the output gate and the updated cell state:(9)$$o_{t}=\sigma \left(x_{t}w_{o}+h_{t-1}{wh}_{o}+{bias}_{o}\right).$$

Here, $o_{t}$ represents the output of the output gate at the current time step, and ${wh}_{o}$ is the weight matrix for the hidden state.

The updated cell state, which passes through Equations (5)–(8), undergoes the sigmoid activation function of Equation (9) above and is converted into a value between 0 and 1, ultimately determining the information to be output:(10)$$h_{t}=o_{t}⊙\tanh\left(c_{t}\right).$$

Equation (10) above shows the hidden state update. It is calculated based on the updated cell state of Equation (8) and the output data of Equation (9). After passing through the tanh, the value falls within the range of −1 to 1. Then, it is multiplied by the output, and the final result is retained in the long-term memory. It represents the output of the LSTM cell for the current time point.

It comprises an input layer, two hidden layers, and an output layer. The input layer has two nodes, the hidden layers have four and two nodes, respectively, and the output layer has one node. In Figure 3, “w” represents the weights of the nodes. The DC power generation and solar radiation selected through the correlation analysis were used as the input values in the model. Regarding the activation function, a linear function was used in order to use the output value calculated through the model as a predicted value. The model’s training was performed on preprocessed data at the ratio of 80:20.

### 3.3. Model Evaluation Method

For this study, a hybrid model combining the CNN-LSTM was built. The common performance metrics used to evaluate the performance of time series data prediction models are the Mean Absolute Error (MAE), the Root Mean Square Error (RMSE), and the Mean Absolute Percentage Error (MAPE). The symbol N used in the quantitative index is the number of test data, $x_{pred}$ is the value predicted through the proposed algorithm, and $x_{act}$ the actual value.

The MAE can be used to measure the error between the predicted and measured values. It depends on the scale of the continuous variable. When a MAE value is lower, it indicates a higher level of accuracy of the prediction model. The definition of the MAE is the same as that provided in Equation (11):(11)$$MAE=\frac{1}{N}\sum_{i=1}^{n}\left|x_{pred\;}-x_{act}\right|.$$

The RMSE measures the difference between the predicted value and the actual value. It can be used to measure the deviation between predicted and measured values. The difference between the RMSE and the MAE is that the RMSE is sensitive to outliers, meaning that it is easily influenced by large deviations. Since the error is squared, the larger the error, the larger the weight reflected. Thus, the closer the RMSE result is to 0, the better the performance is. The RMSE definition is the same as that provided in Equation (12):(12)$$RMSE=\sqrt{\frac{1}{n}\sum_{i=1}^{n}{\left(x_{pred}-x_{act}\right)}^{2}}.$$

The MAPE is an indicator of the extent to which an error accounts for the predicted value. Equation (13) represents the formula for the MAPE. The MAPE is resistant to outliers, but it implies that it is difficult to identify errors intuitively compared to the MAE. The MAP has a percentage value, and it can be interpreted that the closer to zero it is, the better the performance of the predictive model:(13)$$MAPE=\frac{1}{n}\sum_{i=1}^{n}\left|\frac{x_{pred}-x_{act}}{x_{act}}\right|\times 100\%.$$

The predicted generation using the LSTM has errors. The size of the error varies depending on the density of the predicted power generation data. When the density of the predicted values is high, the error between the predicted value and the actual value is small; conversely, if the density is low, the error is large. The size of the error varies depending on the density, and this changes the accuracy of the model. The accuracy of the prediction model is determined by density. *R*^2^ is used to measure the strength of the correlation between the predicted value output from the prediction model and the actual measured value. $R^{2}$ has the range 0 ≤$\;R^{2}$ ≤ 1. The closer it is to 0, the lower the accuracy of the prediction model, whereas the closer it is to 1, the higher the accuracy. Equation (14) represents $R^{2}$.
(14)$$R^{2}=1-\frac{\sum_{i=1}^{n}{\left(x_{pred}-x_{act}\right)}^{2}}{\sum_{i=1}^{n}{\left({\bar{x}}_{act}-x_{act}\right)}^{2}}$$

## 4. Experiment

This section presents the analysis of the decrease in the ratio of the AC power generation compared to the DC power generation using preprocessed data and the proposed LSTM model. Before the experiment, the specifications of the hardware and software used in the experiment were as follows: the CPU was an AMD Ryzen 5 5600X, the GPU was an NVIDIA RTX 2070, and 32GB RAM was used. Data preprocessing was performed using the R programming language, and the models and testing were performed using Python-based PyTorch (version 1.8).

Table 5 shows the quantitative evaluation of the proposed LSTM model derived from the training dataset. It shows that the MAPE was 7.36, the RMSE was 27.91, the MAE was 18.43, and $R^{2}$ was 0.97. Using the trained model, the AC power generation data of inverters operated for the 1st year, the 2nd year, and the 3rd year were predicted. To confirm the decreasing trend in the conversion rate of the AC power generation compared to the DC power generation by year, an analysis was conducted at different time points for each year.

Figure 4a is a graph showing the prediction of power generation from March 2020, the initial operation period, to December 2020, using the proposed algorithm. Figure 4b shows the graph for the second year of operation, from January 2021 to December 2021. Figure 4c shows the graph for the third year of operation, from January 2022 to December 2022. Looking at the trend of the graph through a qualitative evaluation, it can be seen that the predicted value follows the trend of the actual value very well, although some predicted values at the maximum and minimum points generally showed lower values than the actual values. An additional error analysis was performed to accurately identify the relationship between the actual value and the predicted value. The errors between the predicted values derived through the model and the actual power generation values were subjected to a statistical analysis.

In addition, we conducted further experiments for aging verification by introducing RNN (Recurrent Neural Network) and GRU (Gated Recurrent Unit) models to compare their performance with the LSTM. The RNN is a basic recurrent neural network structure that can learn patterns in time series data and is suitable for processing continuous data. However, it may suffer from performance degradation when predicting long sequences due to long-term dependency issues. To address this, the GRU, which employs a gate mechanism, was introduced to improve upon the shortcomings of the RNN. The GRU effectively controls the flow of information and enhances performance, offering a simpler structure and faster learning speed while maintaining similar performance to the LSTM. This study utilized the characteristics of the RNN and the GRU to perform aging verification and evaluate the impact of various time series models on an aging analysis.

Continuing with the results of Figure 4 shown previously, Figure 5 and Figure 6 show the power generation forecast results using the RNN and LSTM models for the same period. Specifically, Figure 5a and Figure 6a show the prediction results for 2020 using the LSTM and RNN models, respectively. Similarly, Figure 5b and Figure 6b show the prediction results for 2021, and Figure 5c and Figure 6c show the results for 2022.

Visually comparing the prediction values of the RNN and LSTM models with the actual power generation data shows that both models are effective in time series forecasting by capturing the overall trend of the data. However, subtle differences can be observed, especially at peak and trough points, where the prediction values sometimes deviate from the actual measurements. The analysis results show that both models follow the overall trend well, but the RNN model sometimes shows slightly larger deviations than the LSTM model, especially when the power generation fluctuates sharply.

Table 6 is a quartile table showing the results of an analysis of the distribution of the error values. Quartiles are boundary values for dividing data equally into four groups: Q1 is the point where 25% of the data are distributed; Q2 is where 50% of the data lie; and Q3 is where 75% of the data fall. The results shown in Table 6 are as follows. It can be seen that the prediction error quartile from 2020 to 2022 is gradually increasing, which means that the error values in 2022 are distributed over a larger range compared to 2020. The average error can also be seen to increase from 410.71 in 2020 to 426.36 in 2021, and to increase further still to 570.26 in 2022. The average prediction error for 2020 is 410.71, but the average error for 2022 is 570.26. Compared to 2020, the error in 2022 increased by an average of 159.55, corresponding to 0.75% of the inverter’s maximum power generation capacity, which is 21 kW.

Figure 7 shows the residuals between the observed data and the predicted data for the yearly data. Here, the dotted line means the section of the standard deviation of the residuals. Figure 7a shows the residual graph for the year 2020, with a standard deviation of 493.60 and a mean of 410.71. Figure 7b shows the residual graph for the year 2021, with a standard deviation of 542.56 and a mean of 426.36. Figure 7c shows the residual graph for the year 2022, with a standard deviation of 652.08 and a mean of 570.26. It can be seen that as the greater the number of operating days, the more values there are outside the standard deviation range. This shows that differences between the predicted and actual measured values gradually occur, showing that the efficiency is decreasing.

Additionally, Figure 8 and Figure 9 show the residuals for the prediction results using the RNN model and the GRU model, respectively. The dotted lines in these graphs represent the standard deviation range of the residuals.

Figure 8 shows the residuals for the prediction results of the RNN model. Figure 8a corresponds to the first year, and we can see that the residuals are relatively evenly distributed around the mean, with some peaks slightly outside the standard deviation range. This means that there is some deviation in the predicted values. As can be seen in Figure 8b,c, the distribution of the residuals becomes wider as we move to the second and third years, and the density outside the standard deviation range increases. This reflects that the predicted values tend to deviate more frequently from the actual measurements over time, and the prediction accuracy gradually decreases.

Similarly, Figure 9 shows the residuals for the prediction results of the GRU model. Figure 9a corresponds to the first year, and the residuals are densely distributed within the standard deviation range around the mean, indicating strong consistency between the predicted and actual values. However, as can be seen in Figure 9b,c, the distribution of residuals becomes wider as the third year progresses, and the values outside the dotted line range increase noticeably. This shows that, like the RNN model, the prediction accuracy of the GRU model also tends to decrease as the operating period increases. Through this additional analysis, we can confirm that both the RNN and GRU models show a tendency for the residuals to increase as the operating period increases, which means that the difference between the predicted value and the actual value is gradually increasing.

We also analyzed the models using the IAE (Integral Absolute Error), the RE (Relative Error), and the SD (Standard Deviation) indices for the years 2020, 2021, and 2022. This allowed us to observe how the error between the prediction model and the actual values changes over time. The results of the the IAE, RE, and SD analyses suggest that performance degradation of inverters and PV modules can occur over time. Therefore, these analyses can be used as a method to predict potential performance degradation and aging, emphasizing the importance of maintenance and management of power generation systems.

Figure 10, Figure 11 and Figure 12 present the analysis results of the IAE, the RE, and the SD for the power generation prediction results using the LSTM, RNN, and GRU models.

To analyze the predictive performance of each model more specifically, the validation results from 2020 to 2022 are shown in Table 7.

In 2020, the LSTM model recorded a MAPE of 7.68, an RMSE of 511.58, and a MAE of 402.88, demonstrating superior results compared to the RNN and GRU models. The RNN model showed a MAPE of 7.98, an RMSE of 481.37, and a MAE of 384.15; while it had the lowest RMSE and MAE values, its MAPE was slightly higher than that of the LSTM. The GRU model recorded a MAPE of 7.87, an RMSE of 519.93, and a MAE of 416.16, exhibiting performance between the LSTM and RNN models, but it had the highest RMSE value. In 2021, the LSTM model achieved a MAPE of 8.44, an RMSE of 516.85, and a MAE of 404.88. In contrast, the RNN model maintained the lowest values with an RMSE of 484.24 and a MAE of 384.98, showing consistent predictive performance. The GRU model produced somewhat unstable results with a MAPE of 8.50, an RMSE of 526.06, and MAE of 419.09. In 2022, a performance decline was observed across all three models. Specifically, the RMSE values for the LSTM and GRU models increased significantly compared to 2020 and 2021, suggesting a drop in performance for long-term data predictions. Meanwhile, the RNN model still showed relatively stable performance in the MAE and RMSE, but the values increased compared to 2020 and 2021.

This performance analysis of predictive models can serve as important data for preemptively determining the obsolescence of power generation systems. By comparing the characteristics of each LSTM, RNN, and GRU model, it is possible to detect failures or efficiency degradation in power generation systems over time based on the observed performance changes.

## 5. Conclusions

This paper attempted to diagnose the deterioration of inverter efficiency by using power generation data and environmental sensor data collected from inverters with different manufacturing dates (years). This study identified degradation trends in inverter efficiency and confirmed the value of the decrease in efficiency by conducting a regression analysis of DC and AC power generation data. In order to analyze changes in efficiency according to the extent of deterioration, a time series data-based LSTM power generation prediction model was implemented. The data used for creating the model were obtained from inverters and environmental sensors located at a photovoltaic power plant that has been operational since January 2020. To ensure data consistency, the preprocessing task of interpolating outliers and missing values was applied to the data and reconstructed into a data range of between -1 and 1 through normalization. Then, an AC power generation prediction model was used to analyze the drop in efficiency based on these preprocessed data. The power generation prediction model receives DC power generation and solar radiation data and derives the results using the LSTM. In general, various parameters and additional modules are utilized to increase the accuracy of data prediction, but this requires an additional data collection process and hence becomes a factor that causes overfitting.

As a result of verifying the proposed algorithm with a quantitative evaluation index, the MAPE was derived as 7.36, which is 92.64% when converted into a percentage. The verified model was then applied to the inverter data to analyze the decrease in efficiency. The data used for this analysis are those obtained from the 1st, 2nd, and 3rd year after the inverter started operating in 2020. Each time point datum was compared by calculating the error between the predicted power generation derived by the predictive model and the actual value. Through this comparison, it could be seen that the trend of the predicted value and the actual value was similar, although the predicted value was generally lower than the actual value. The residuals for data predictions in the first year of operation showed an average range of 410.71 based on the actual value; the residuals for data predictions in the second year of operation showed an average range of 426.36 compared to the actual values; and, lastly, the residuals for data predictions in the third year of operation showed a range of 570.26, representing an increase of 159.55 compared to the first year. The reason for this can be described as follows. The Republic of Korea shows regular seasonal cycles and patterns, such as spring, summer, fall, and winter. However, meteorological conditions, such as weather, temperature, precipitation, and solar radiation throughout the year, are determined by various factors that affect the seasons. Thus, the fact that the same meteorological conditions were not observed between the first and second years is attributable to the various variables and uncertainties related to nature. For this reason, it is not easy to assume or guarantee the same pattern of weather conditions from year to year. It is difficult to guarantee that the type of power generation data affected by the weather environment will be the same each year. In the case of environmental data, the solar radiation data, excluding temperature, show high values on clear days and low values on cloudy days, so they are less affected by seasonality. Even if the solar radiation values at a given time of the year vary from year to year, the predicted value should be output in a similar form for similar input values. However, the derived results show that the results differed as the years of operation progressed.

The results of this research indicate that the actual power generation of the diagnosed inverter decreased by 0.7% compared to the predicted power generation over three years. These findings suggest a gradual decline in conversion efficiency due to the degradation of the panel or inverter over time. Assuming that the inverter’s conversion efficiency decreases equally each year, these results can help identify the pattern of efficiency decline over time. Moreover, when the inverter’s power generation is lower than expected, the proposed model can be utilized as a valuable method among various diagnostic approaches to help determine whether this phenomenon is due to a decrease in efficiency caused by the aging of the inverter.

## Figures and Tables

**Figure 1 sensors-24-06390-f001:**
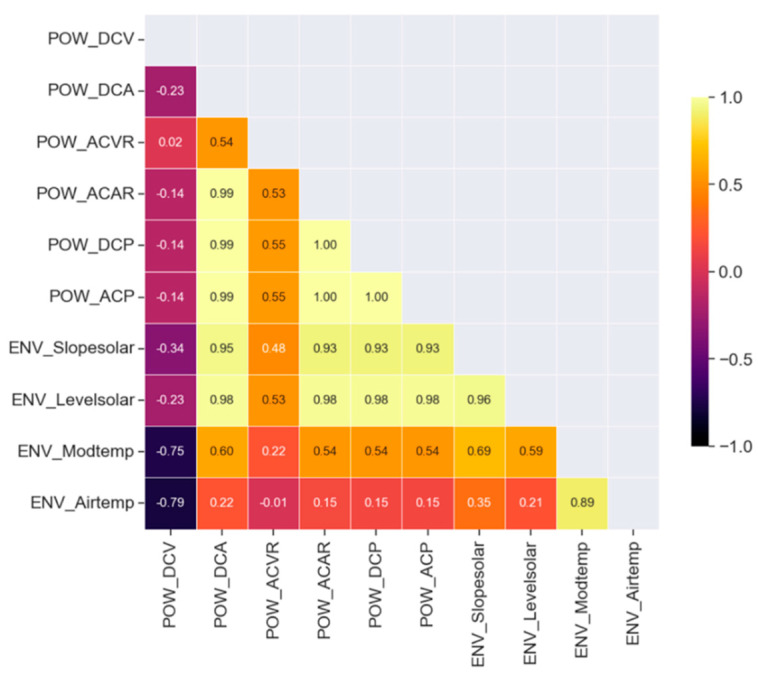
Correlation matrix analyzing the correlation between power generation data.

**Figure 2 sensors-24-06390-f002:**
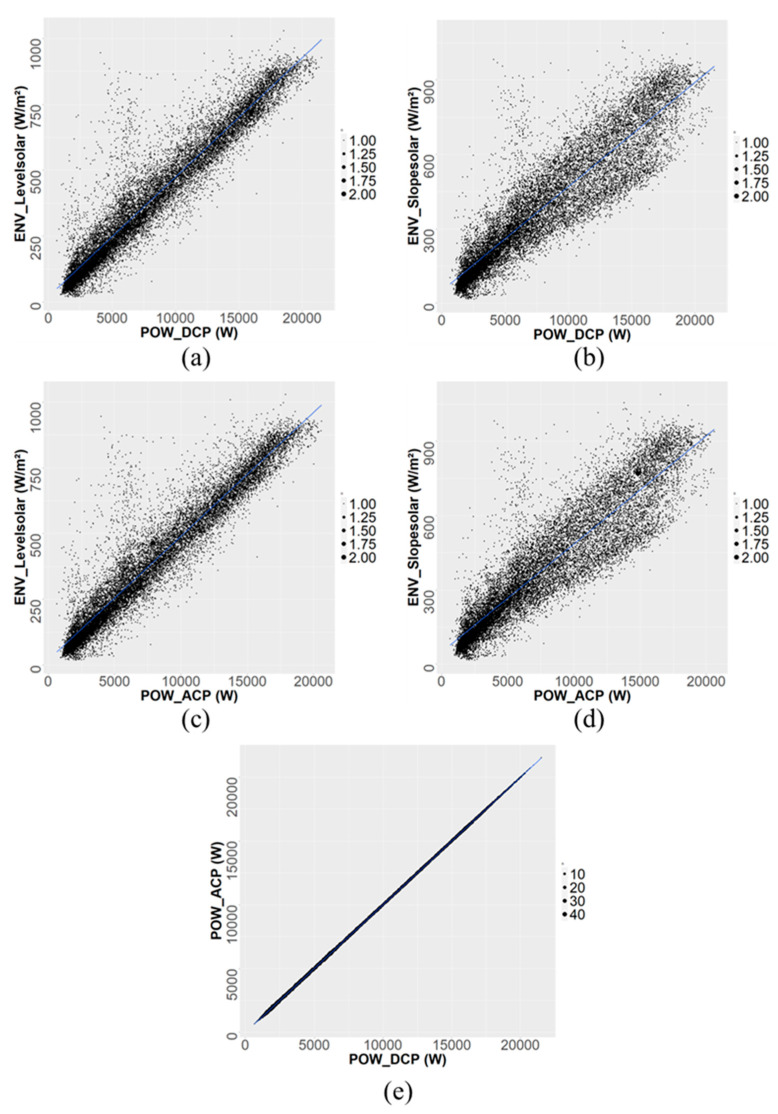
Scatter plot of a linear correlation analysis between two variables: (**a**) POW_DCP and Levelsolar; (**b**) POW_DCP and ENV_Slopesolar; (**c**) POW_ACP and ENV_Levelsolar; (**d**) POW_ACP and ENV_Slopesolar; and (**e**) POW_DCP and POW_ACP.

**Figure 3 sensors-24-06390-f003:**
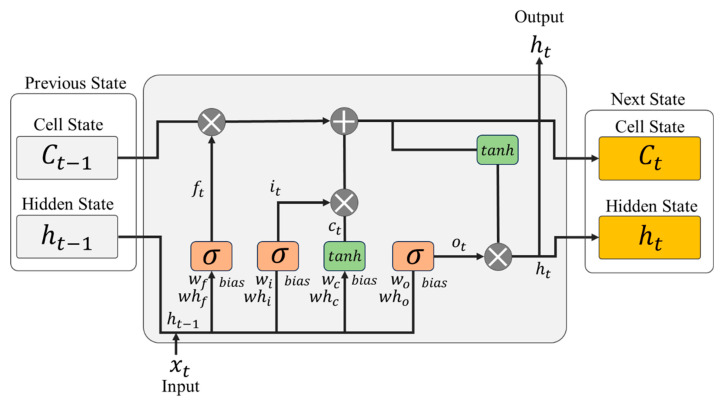
Cell Diagram of the Long Short-Term Memory (LSTM).

**Figure 4 sensors-24-06390-f004:**
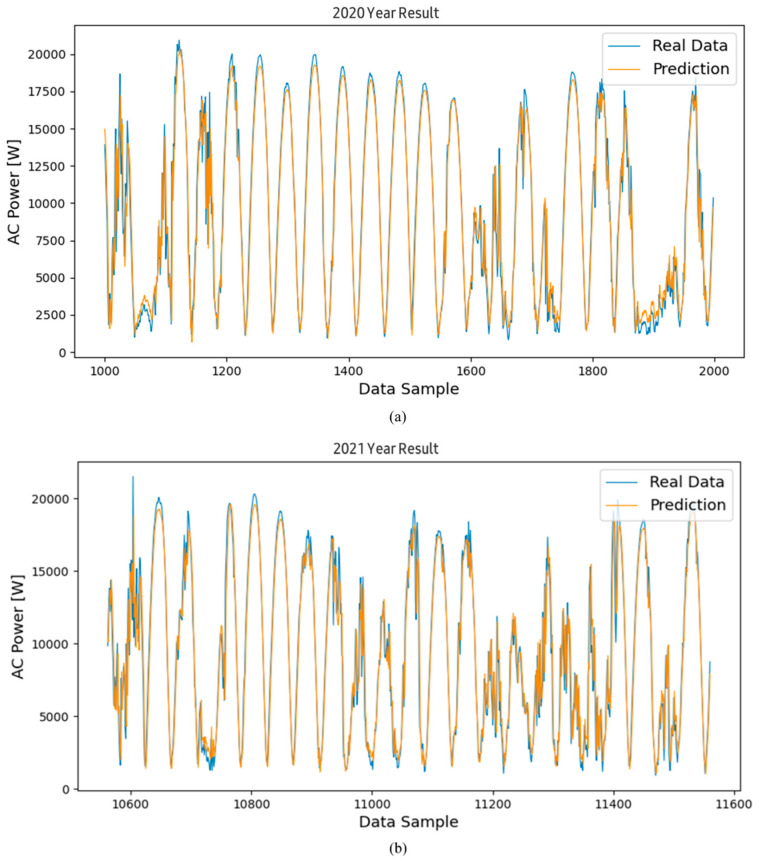
Results of the predicted and actual measured values of the inverter’s AC power using the LSTM model; graphs of the inverted operated for (**a**) data for the 1st year, (**b**) data for the 2nd year, and (**c**) data for the 3rd year.

**Figure 5 sensors-24-06390-f005:**
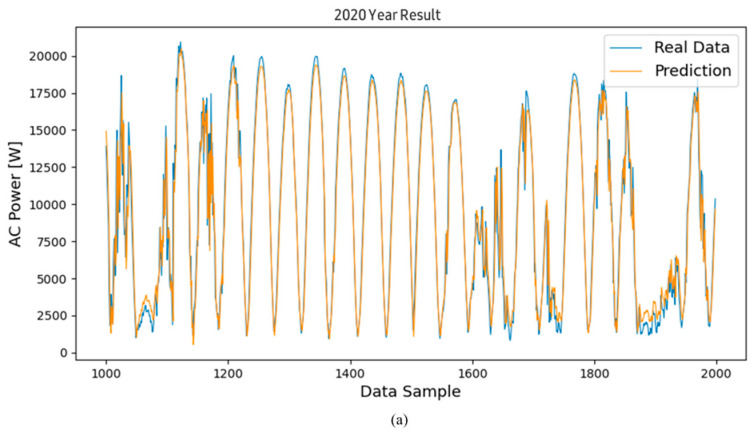
Results of the predicted and actual measured values of the inverter’s AC power using the RNN model; graphs of the inverted operated for (**a**) data for the 1st year, (**b**) data for the 2nd year, and (**c**) data for the 3rd year.

**Figure 6 sensors-24-06390-f006:**
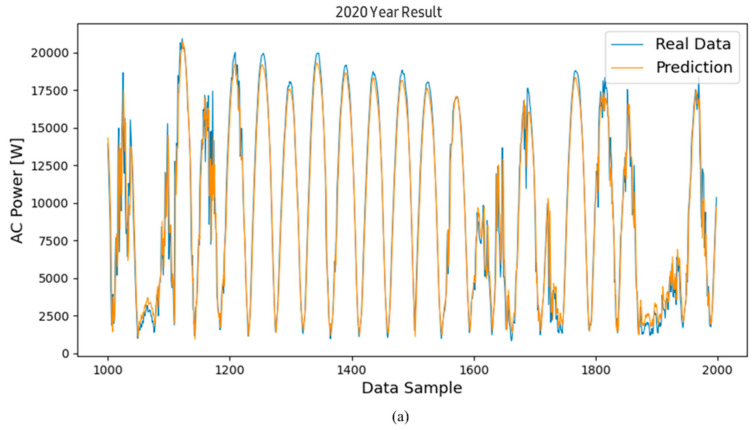
Results of the predicted and actual measured values of the inverter’s AC power using the GRU model; graphs of the inverted operated for (**a**) data for the 1st year, (**b**) data for the 2nd year, and (**c**) data for the 3rd year.

**Figure 7 sensors-24-06390-f007:**
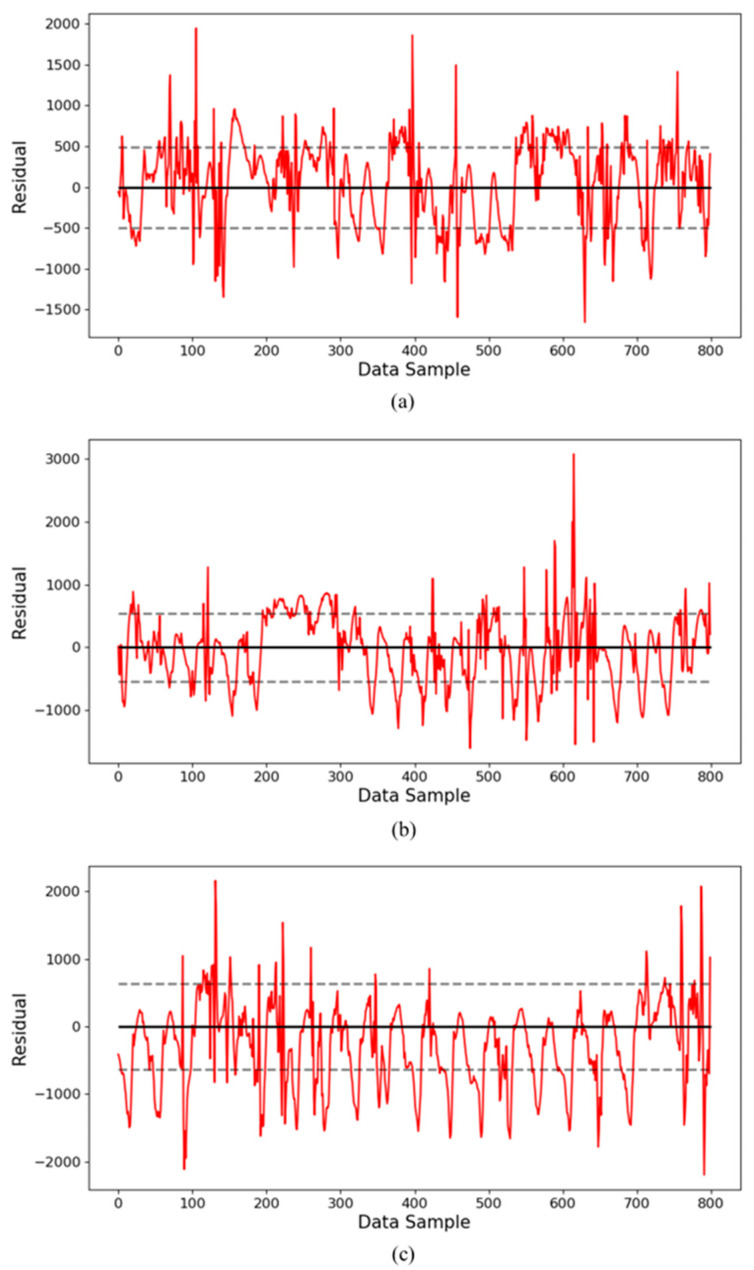
Residual graphs of prediction results by year using the LSTM model: residuals of inverter operated for (**a**) data for the 1st year, (**b**) data for the 2nd year, and (**c**) data for the 3rd year.

**Figure 8 sensors-24-06390-f008:**
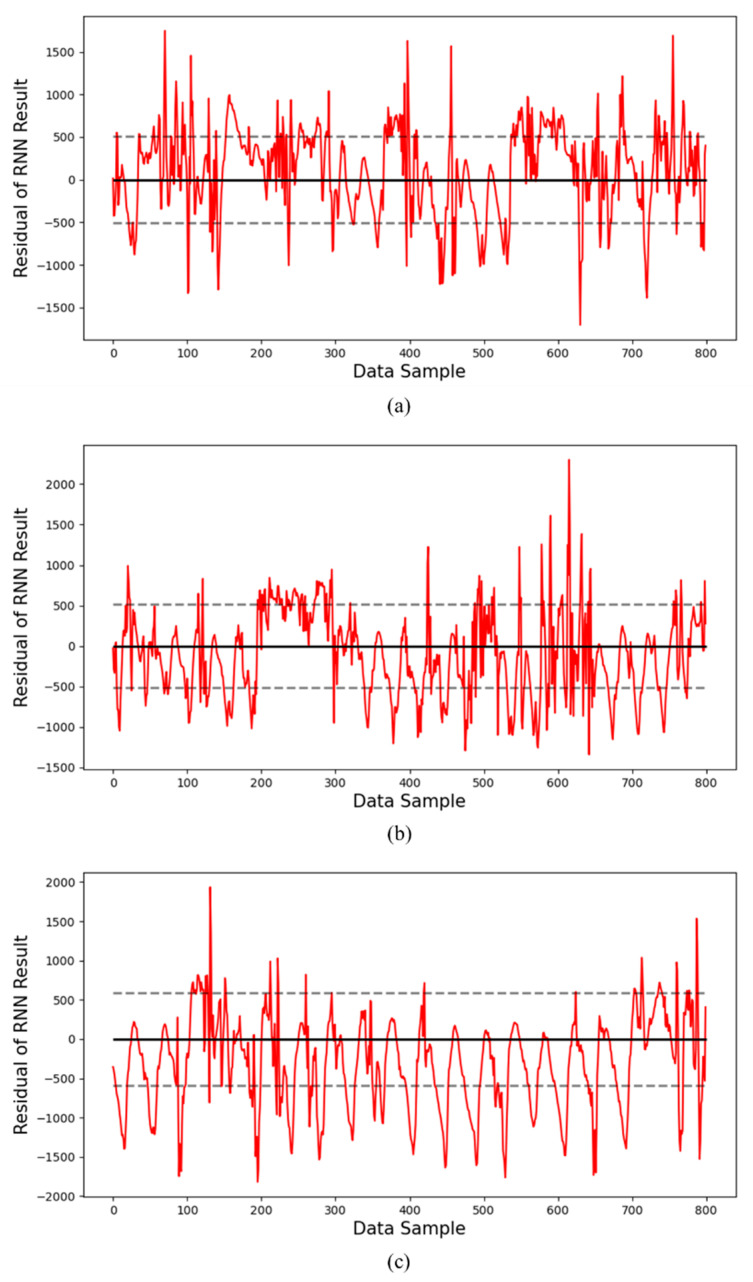
Residual graphs of prediction results by year using the RNN model: residuals of the inverter operated for (**a**) data for the 1st year, (**b**) data for the 2nd year, and (**c**) data for the 3rd year.

**Figure 9 sensors-24-06390-f009:**
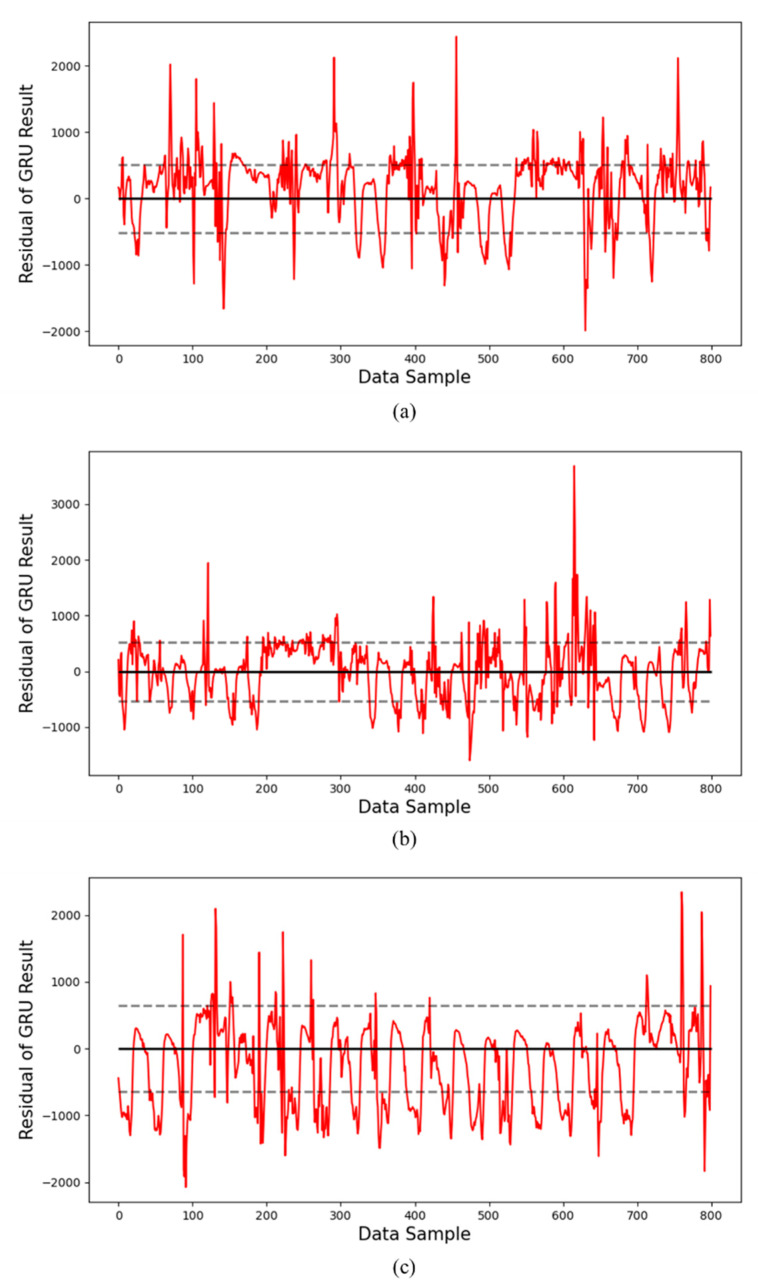
Residual graphs of prediction results by year using the GRU model: residuals of inverter operated for (**a**) data for the 1st year, (**b**) data for the 2nd year, and (**c**) data for the 3rd year.

**Figure 10 sensors-24-06390-f010:**
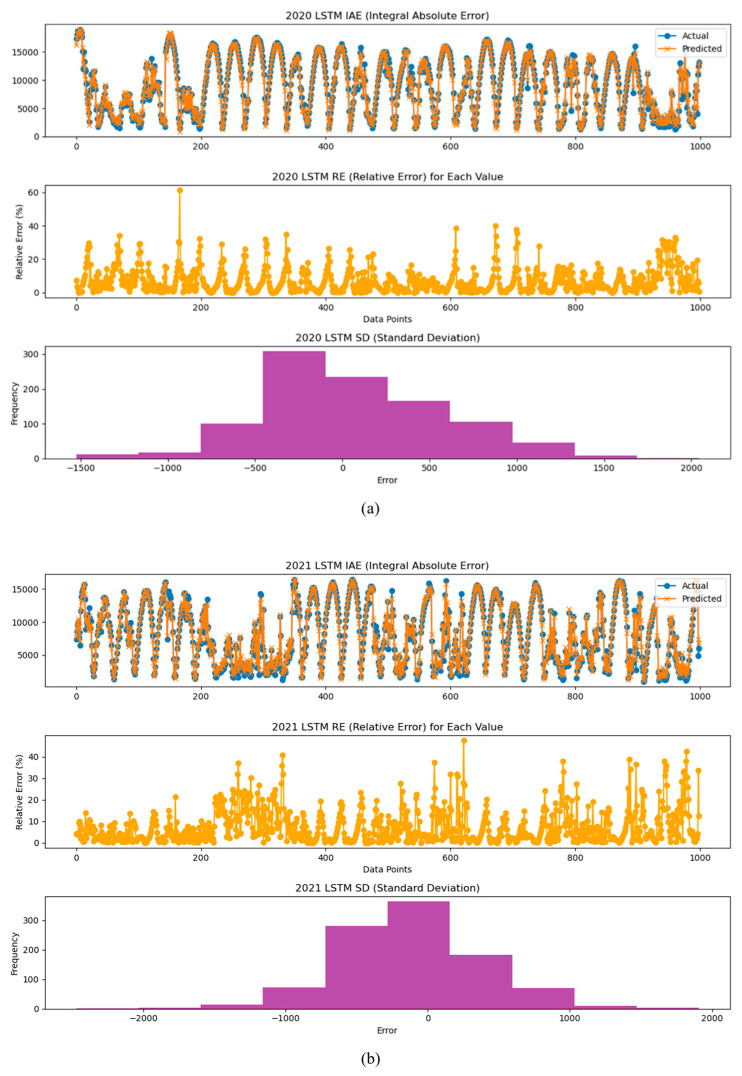
Analysis graphs of the IAE, RE, and SD of prediction results by year using the LSTM model: analysis of inverter performance based on (**a**) 2020-year data, (**b**) 2021-year data, and (**c**) 2022-year data.

**Figure 11 sensors-24-06390-f011:**
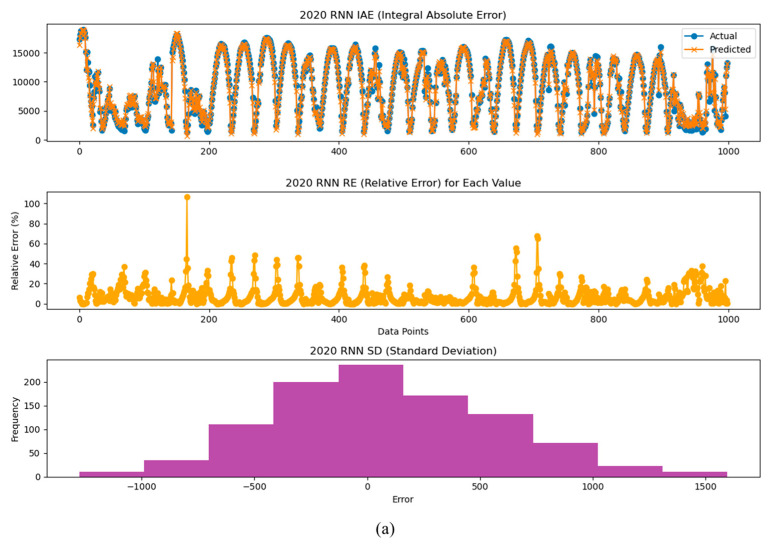
Analysis graphs of the IAE, RE, and SD of prediction results by year using the RNN model: analysis of inverter performance based on (**a**) 2020-year data, (**b**) 2021-year data, and (**c**) 2022-year data.

**Figure 12 sensors-24-06390-f012:**
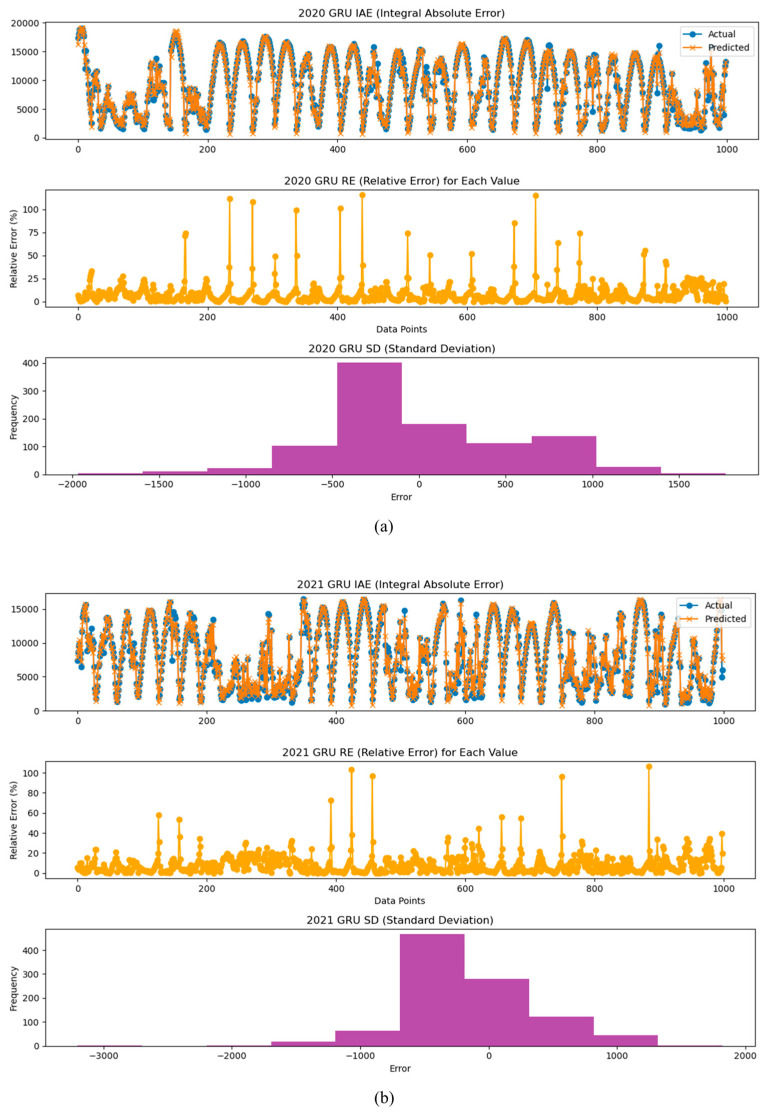
Analysis graphs of the IAE, RE, and SD of prediction results by year using the GRU model: analysis of inverter performance based on (**a**) 2020-year data, (**b**) 2021-year data, and (**c**) 2022-year data.

**Table 1 sensors-24-06390-t001:** Environment data schema for the photovoltaic monitoring system.

Column	Data Type	Default	Description
ENV_DATE	DATE	NULL	DATE
ENV_TIME	TIME	NULL	TIME
ENV_Slopesolar	FLOAT	NULL	Horizontal Solar Radiation
ENV_Levelsolar	FLOAT	NULL	Vertical Solar Radiation
ENV_Modetemp	FLOAT	NULL	Module Temperature
ENV_Airtemp	FLOAT	NULL	Outside Temperature

**Table 2 sensors-24-06390-t002:** Inverter data schema for photovoltaic monitoring system.

Column	Data Type	Default	Description
POW_DATE	DATE	NULL	Collected DATE
POW_TIME	TIME	NULL	Collected TIME
POW_DCV	INT	NULL	DC Voltage
POW_DCA	DOUBLE	NULL	DC Ampere
POW_ACVR	DOUBLE	NULL	AC Voltage
POW_ACAR	INT	NULL	AC Ampere
POW_DCP	INT	NULL	DC Power (W)
POW_ACP	INT	NULL	AC Power (W)

**Table 3 sensors-24-06390-t003:** Elementary statistics for photovoltaic system data.

Data (Unit)	Min.	Median	Mean	Max.	Standard Deviation
**POW_DCV**	314.5	553.8	555.1	675.7	30.4
**POW_DCA**	1.7	13.3	15.5	39.7	9.7
**POW_ACVR**	193	396	396.1	416.7	5.8
**POW_ACAR**	2	11.2	12.7	31	7.2
**POW_DCP(W)**	660	7291	8423	22,500	5303
**POW_ACP(W)**	653	7045.5	8134.5	21,512.5	5097.8
**ENV_Slopesolar** ** $\;(W/m^{2})$ **	17.2	361.7	400.6	1088.5	239.5
**ENV_Levelsolar** ** $\left(W/m^{2}\right)$ **	20.33	360.1	398.8	1028.2	246.1
**ENV_Modetemp**	−18.1	27.88	27.32	62.15	13
**ENV_Airtemp**	−16.2	20.84	19	40	10

**Table 4 sensors-24-06390-t004:** Correlation coefficients between power and solar radiation.

Corr.	Vertical Solar Radiation	Horizontal Solar Radiation
**DC Power**	0.9257	0.9762
**AC Power**	0.9276	0.9781

**Table 5 sensors-24-06390-t005:** Results of validation of the estimation model.

	MAPE	RMSE	MAE	$R^{2}$
Result	7.36	27.91	18.43	0.97

**Table 6 sensors-24-06390-t006:** Quartile results for inverter error analysis.

	Q1 (25%) [W]	Q2 (50%) [W]	Q3 (75%) [W]	Average Error
Data of 1st year (2020)	−315.26	142.02	406.90	410.71
Data of 2nd year (2021)	−393.77	−13.94	338.21	426.36
Data of 3rd year (2022)	−757.39	−289.27	151.62	570.26

**Table 7 sensors-24-06390-t007:** Validation results based on the test dataset using the LSTM, RNN, and GRU models.

YEAR	MODEL	MAPE	RMSE	MAE
2020	LSTM	7.68	511.58	402.88
	RNN	7.98	481.37	384.15
	GRU	7.87	519.93	416.16
2021	LSTM	8.44	516.85	404.88
	RNN	8.60	484.24	384.98
	GRU	8.50	526.06	419.09
2022	LSTM	8.16	528.63	412.78
	RNN	8.57	495.45	394.80
	GRU	8.36	536.90	420.23

## Data Availability

Data is contained within the article.

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
