# Peer review of "Analysis of Inverter Efficiency Using Photovoltaic Power Generation Element Parameters"

_sensors, 2024, doi:10.3390/s24196390_

Round 1

Reviewer 1 Report

Comments and Suggestions for Authors

The article proposes a model to determine and predict the efficiency loss of inverters associated with photovoltaic systems.

For this purpose, the authors build a LTSM model aided by neural networks trained by means of irradiance data sets in the electrical and panel irradiance that, after verification, they declare capable of analyzing and predicting such efficiency loss throughout the lifetime of the inverters in photovoltaic plants.

The authors present background and methodology for the preparation of the model and experimental results on which the above assertions are based. They also rely on a good number of bibliographic references with a high rate of actuality.

I will now list a series of comments with the aim of increasing, if possible, the quality of the proposal:

- In Fig. 2 a numerical gradation of dot thickness appears to the right of the subfigures that is not referred to or explained in the caption or in the text.

- Regarding model creation and verification, section 3.2, explaining the LSTM cell diagram from line 343 to 437 requires a thorough review to give coherence and consistency to the explanation of the methodology developed. And, unless I have made a mistake, let me explain:

o The explanation of formulas (5) to (10) refer to variables and parameters that, although said to be shown, do not appear in Fig. 3, i.e. in line 411 it refers to the weight 'w' of the nodes that appears in the figure without this being the case.

o The lines that indicate the flow of variables do not all have the sense of operation do not explicitly incorporate neither the blocks of operation nor the operands that relate those six equations.

o In equations (8) and (10) a mathematical operator (asterisk in a circle) appears of which I do not know its meaning (I apologize in advance) because my search indicates that it is a special neutral element binary operation while in line 406 it indicates 'multiplied'...- Regarding the model evaluation, section 4.1, since it explains the validation method, it could also be considered opportune to place it at the end of section 3 instead of at the beginning of section 4.

- The captions of subfigures 4 refer to 'years' when they are only for a single year.

- The captions of figures 4 and 5 refer to 1, 2, and 3 years when the body of the text and table 6 refer to the first, second, and third years. The same happens in line 474.

- The annotated data in Table 6 (lines 496 to 506) referring to the mean error (lines 502 and following) indicate numerical data that I cannot find in the table. The same happens in the conclusions (lines 549 et seq.) - In line 506, does it refer to 21 kW? In addition, the international system requires a non-separation space between the magnitude and the units.

- The statement (lines 573-575) that it can be concluded that the proposed model can identify whether converter efficiency loss can be associated with aging, I believe, is not sufficiently supported by the results. In fact, it is not an assertive conclusion, it is a conditional sentence.

Some other formal aspects associated with the template are:

- In lines 35, 69, 70, 79, 79, 84, 87, 87, 123, 128, 131, 131, 146, 152, 152, 154, 157, 157, 165, 165, 167, 170, 171, 171, 176, 180, 182, and 184 there seems to be a space missing before the opening square brackets of the quote.

- In lines 49 and 63 there is a period just before the opening square brackets of the quote instead of a space.

- Tables 1 and 3 are separated by two pages. Neither should Fig. 4.

- The formula for the equations should be centered in the paragraph.

Author Response

Please, refer to the attached file

Thank you!

Reviewer 2 Report

Comments and Suggestions for Authors

the paper is written 

but need an improvement like, in the state of the art ad a section about the forecasting categories, short term forecasting, nowcasting, medium, long ....

and add a paragraph about the method of forecasting, physical methods, statistical method, and hybrid method with a description you can find it at

https://doi.org/10.1016/j.renene.2017.05.063

https://doi.org/10.1016/j.ijleo.2020.165207

the state of the art in introduction must be from various techniques and methods not only one method and from the recent used methods

in results section

calculate the IAE, RE, SD, and plot the RE and IAE in figures with a comparison  ith other methods from physical methods, statistical method and hybrid methods

Author Response

Please, refer to the attached reply file!

Thank you!

Round 2

Reviewer 2 Report

Comments and Suggestions for Authors

the equation 4 present e data were normalized within the range of -1.0 to 1.0.

why use it? normally in the results must be between -1 and 1.

results in Figures 4, 5 and 6  is not compared with other methods

you must compare with other methods and techniques to validate your results and to conclude about your method.

we can't know your method is better only if you compare with min 2 or 3methods
